# The *Plasmodium falciparum* rhoptry bulb protein RAMA plays an essential role in rhoptry neck morphogenesis and host red blood cell invasion

Emma S. Sherling[1], Abigail J. Perrin[1], Ellen Knuepfer[2], Matthew R. G. Russell[3], Lucy M. Collinson[3], Louis H. Miller[4], Michael J. Blackman[1,5]*

1 Malaria Biochemistry Laboratory, The Francis Crick Institute, London, United Kingdom, 2 Malaria Parasitology Laboratory, The Francis Crick Institute, London, United Kingdom, 3 Electron Microscopy Science Technology Platform, The Francis Crick Institute, London, United Kingdom, 4 Laboratory of Malaria and Vector Research, National Institute of Allergy and Infectious Diseases, National Institutes of Health, Rockville, Maryland, United States of America, 5 Faculty of Infectious and Tropical Diseases, London School of Hygiene & Tropical Medicine, London, United Kingdom

* mike.blackman@crick.ac.uk

**Data Availability Statement:** All relevant data are within the manuscript and its Supporting Information files.

## Abstract

The malaria parasite *Plasmodium falciparum* invades, replicates within and destroys red blood cells in an asexual blood stage life cycle that is responsible for clinical disease and crucial for parasite propagation. Invasive malaria merozoites possess a characteristic apical complex of secretory organelles that are discharged in a tightly controlled and highly regulated order during merozoite egress and host cell invasion. The most prominent of these organelles, the rhoptries, are twinned, club-shaped structures with a body or bulb region that tapers to a narrow neck as it meets the apical prominence of the merozoite. Different protein populations localise to the rhoptry bulb and neck, but the function of many of these proteins and how they are spatially segregated within the rhoptries is unknown. Using conditional disruption of the gene encoding the only known glycolipid-anchored malarial rhoptry bulb protein, rhoptry-associated membrane antigen (RAMA), we demonstrate that RAMA is indispensable for blood stage parasite survival. Contrary to previous suggestions, RAMA is not required for trafficking of all rhoptry bulb proteins. Instead, RAMA-null parasites display selective mislocalisation of a subset of rhoptry bulb and neck proteins (RONs) and produce dysmorphic rhoptries that lack a distinct neck region. The mutant parasites undergo normal intracellular development and egress but display a fatal defect in invasion and do not induce echinocytosis in target red blood cells. Our results indicate that distinct pathways regulate biogenesis of the two main rhoptry sub-compartments in the malaria parasite.

## Author summary

Despite improved control measures over recent decades, malaria is still a considerable health burden across much of the globe. The disease is caused by a single-celled parasite

**Funding:** ESS was in receipt of a Wellcome Trust/
National Institutes of Health PhD studentship
(103459/Z/14/Z), whilst AJP was supported by
Wellcome Trust grant 106239/Z/14/A (to MJB).
Work in the LHM laboratory was supported by the
Intramural Research Program of the Division of
Intramural Research, National Institute of Allergy
and Infectious Diseases, NIH. This work was
supported by funding to MJB from the Francis
Crick Institute (https://www.crick.ac.uk/) which
receives its core funding from Cancer Research UK
(FC001043; https://www.cancerresearchuk.org),
the UK Medical Research Council (FC001043;
https://www.mrc.ac.uk/), and the Wellcome Trust
(FC001043; https://wellcome.ac.uk/). The work
was also supported by Wellcome Trust ISSF2
funding to the London School of Hygiene &
Tropical Medicine. The funders had no role in study
design, data collection and analysis, decision to
publish, or preparation of the manuscript.

**Competing interests:** The authors have declared
that no competing interests exist.

that invades and replicates within host cells. During invasion, the parasite discharges a set
of flask-shaped secretory organelles called rhoptries, the contents of which are crucial for
invasion as well as for modifications to the host cell that are important for parasite sur-
vival. Rhoptry discharge occurs through fusion of the relatively elongated rhoptry neck to
the apical surface of the parasite. Different proteins reside within the bulbous rhoptry
body and the neck regions, but how these proteins are selectively sent to their correct sub-
compartments within the rhoptries and how the rhoptries are formed, is poorly under-
stood. Here we show that a malaria parasite rhoptry bulb protein called rhoptry-associated
membrane antigen (RAMA) plays an essential role in rhoptry neck formation and correct
trafficking of certain rhoptry neck and bulb proteins. Parasites deficient in RAMA pro-
duce malformed rhoptries and–probably as a result—cannot invade host red blood cells.
Our work sheds new light on how rhoptries are formed and reveals insights into the
mechanism by which the correct sorting of proteins to distinct regions of the rhoptry is
regulated.

## Introduction

Malaria is a devastating disease of tropical and subtropical regions. Requiring a mammalian
host and a mosquito vector for transmission, at least six species of the genus *Plasmodium*
cause disease in humans, with *Plasmodium falciparum* being responsible for the great majority
of mortality. All the manifestations of clinical disease result from repeated cycles of invasion,
replication within and lytic egress from red blood cells (RBC). Invasion is an orchestrated pro-
cess, comprising several steps including merozoite attachment, deformation of the RBC mem-
brane, merozoite reorientation, formation of a high affinity interaction between the apical
zone of the merozoite and the RBC surface, active entry, and finally sealing of the RBC mem-
brane behind the intracellular parasite [1–3]. Invasion is generally immediately followed by a
period of transient RBC echinocytosis, a morphological transformation of the RBC surface
into an undulated or 'spiky' appearance, although this can also be induced under certain con-
ditions even in the absence of successful invasion [3–5]. Entry into the host cell occurs con-
comitantly with formation of a membrane-bound parasitophorous vacuole (PV) within which
the invading parasite comes to rest. The parasite then transforms within minutes into a 'ring'
stage form before initiating intracellular development, progressing through a mononuclear
trophozoite stage to a multinucleated schizont which undergoes segmentation to form a new
generation of daughter merozoites. Parasite-induced rupture of the PV membrane (PVM) and
host RBC membrane eventually enables egress of the merozoites to initiate a fresh erythrocytic
cycle.

Egress and invasion involve the regulated discharge of at least four classes of secretory
organelles–rhoptries, micronemes, exonemes and dense granules—that are unique to apicom-
plexan parasites and that are positioned within the apical end of the merozoite [6]. The largest
of these organelles, the rhoptries, are twinned club or pear-shaped structures that are formed
*de novo* during each blood-stage life cycle by progressive fusion of post-Golgi vesicles, with the
possible involvement of an endosome-like pathway [7–9]. Rhoptries are characterised by a rel-
atively wide bulb region which narrows to a neck or duct towards the apical prominence of the
merozoite; rhoptry discharge takes place via the duct. Despite the bulb and neck regions not
being separated by a discernible membranous boundary, examination by transmission elec-
tron microscopy (TEM) has shown a distinct sub-compartmentalisation of each rhoptry, with
the bulb and neck regions displaying different staining characteristics in both *Plasmodium* [7]

and the related apicomplexan parasite *Toxoplasma gondii* [10]. Based on the detailed examination of the fate of a subset of rhoptry neck proteins (RONs; [11]) and bulb proteins, it has been suggested that the contents of the different rhoptry sub-compartments are released at different points in the invasion pathway, with the RONs being discharged before rhoptry bulb proteins [12]. Consistent with this, several RON proteins have been extensively implicated in the early stages of host cell entry, notably in formation of the moving junction or tight junction (TJ), a short-lived doughnut-shaped structure through which the parasite passes as it enters the host cell (e.g. [13–17]) Some members of the *P. falciparum* reticulocyte binding protein homologue (RH) family of RBC adhesins, which also appear to localise to both the rhoptry bulb and neck [18], also play central roles in early steps in invasion (e.g. [3, 19, 20]). However, the association between function, timing of discharge and compartmentalisation within the rhoptries may not be a strict one, since the *Plasmodium*-specific protein RON12 is predominantly released post-invasion, into the nascent PV [21]. Similarly, a rhoptry bub location does not preclude roles during invasion, since the rhoptry bulb protein RhopH3, a component of the high molecular weight (HMW) RhopH complex, is important for RBC entry [22]. In contrast, RhopH3 and its partner proteins RhopH1/Clag and RhopH2 have additionally been shown to be involved in nutrient uptake during development of the intracellular parasite [22–25], whilst the rhoptry bulb low molecular weight (LMW) RAP1/RAP2 complex has been implicated in PVM formation [26]. In *Toxoplasma* (but not in *Plasmodium*) several rhoptry bulb proteins are enzymes, including proteases, phosphatases and kinases or pseudokinases, the latter two groups of which dramatically modulate host cell STAT signalling and the immunity-related GTPase (IRG) pathway involved in controlling parasite replication, and play key roles in virulence [27, 28]. Collectively, the current evidence points to multiple diverse roles for rhoptry proteins, with a general theme being that rhoptry neck proteins are often but not exclusively involved in host cell entry whilst rhoptry bulb proteins usually fulfil subsequent roles in the life cycle (see [29–31] for excellent reviews of this subject). Despite these many insights, of the ~60 known and putative *Plasmodium* rhoptry proteins identified through a combination of numerous antibody studies and proteomic analyses of isolated apicomplexan rhoptries [11, 32–34], the molecular functions of very few *Plasmodium* rhoptry proteins have been characterised. Crucially, how the different subclasses of *Plasmodium* rhoptry proteins are selectively delivered to their distinct sub-compartments within the organelle is unknown.

Rhoptry-associated membrane antigen (RAMA) was initially identified in *P. falciparum* as a rhoptry bulb-resident protein which is expressed relatively early in the asexual blood stage cycle, before the *de novo* formation of nascent rhoptries [35, 36]. Like many other *Plasmodium* rhoptry bulb proteins, RAMA has no obvious orthologue in *Toxoplasma* or other coccidian Apicomplexa. Unique amongst known *Plasmodium* rhoptry proteins, RAMA is associated with the inner (lumenal) face of the rhoptry bulb membrane via a glycosyl phosphatidylinositol (GPI) membrane anchor [36]. Trafficking of RAMA to the rhoptries shares several features of other rhoptry proteins, including transport via the Golgi apparatus and sensitivity to brefeldin A. The ~170 kDa RAMA precursor then undergoes proteolytic processing in the rhoptries or nascent rhoptries, involving removal of an N-terminal segment to generate a mature ~60 kDa form (p60). To gain insights into its function, Topolska and colleagues [36] and Richard et al. (2009) [37] used fluorescent resonance energy transfer (FRET) and immunoprecipitation to demonstrate that RAMA appears to interact with both RhopH3 and RAP1, proteins which are enriched in lipid rafts but that lack lipid anchors. The authors hypothesised that RAMA plays an escorter role in recruiting and trafficking these proteins to the developing rhoptries, and that proteolytic cleavage of RAMA then facilitates their dissociation from the escorter complex. RAMA p60 (but not the RAMA precursor) was detectable in free extracellular merozoites and evidence was also presented that the protein is released from rhoptries at invasion,

supported by the observation that a recombinant form of the extreme C-terminal region of RAMA could bind the RBC membrane. However, in newly invaded 'ring' stage parasites, RAMA p60 was found associated with the PVM, indicating that at least some of the protein was transported into the host cell with the invading parasite [36]. Very recent work has suggested that in fact the RAMA-RAP1 complex may itself be cargo for the *Plasmodium* orthologue of sortilin, an integral membrane protein that in mammalian cells acts as an escorter to transport proteins to the lysosomes, endocytic pathway and plasma membrane [38]. This is consistent with a role in rhoptry formation, since in *Toxoplasma* sortilin is required for biogenesis of both rhoptries and a second class of secretory organelle called micronemes [39], and the *P. falciparum* sortilin orthologue has been localised to the Golgi [9]. Although restricted to the *Plasmodium* genus, *RAMA* orthologues were identified in all *Plasmodium* species examined [40], and this was subsequently confirmed by the now extensive genome sequence data from numerous additional *Plasmodium* species (see PlasmoDB *RAMA* gene ID: PF3D7_070 7300; https://plasmodb.org/plasmo/). More recent evidence from targeted or global reverse genetics studies indicates that disruption of the *P. falciparum* or *P. berghei RAMA* gene is deleterious or lethal [41–43], consistent with an important role for RAMA in the asexual blood stage parasite life cycle. However, the molecular function of RAMA has remained obscure.

Here we have used a robust conditional mutagenesis approach to examine the function of RAMA in the *P. falciparum* asexual blood stage life cycle. In contrast to previous suggestions, we found that disruption of RAMA expression does not affect trafficking of the rhoptry bulb proteins RhopH3 and RAP1, nor biogenesis of the rhoptry bulb. Instead, our results indicate a selective role for RAMA in formation of the rhoptry neck structure. Perhaps as a result, the RAMA mutants show mislocalisation of several RON proteins and a lethal defect in host RBC invasion.

## Results

### Efficient conditional disruption of the *P. falciparum RAMA* gene

Previous attempts to directly disrupt the *P. falciparum RAMA* gene using conventional targeted genetic techniques were unsuccessful [41], suggesting an important role for RAMA in the haploid asexual blood stages. To gain insights into the molecular function of RAMA we therefore used a conditional gene disruption approach based on the rapamycin-inducible DiCre conditional recombinase system. To enable this, we first modified the *RAMA* gene in the DiCre-expressing 1G5DC *P. falciparum* line [44] to introduce 'silent' *loxP* motifs suitable to act as sites for DiCre-mediated site-specific recombination. The *RAMA* gene encodes an 861 amino acid residue protein and comprises 4 small exons plus a single much larger exon (exon 2). The complexity and relatively large size (~3.2 kb) of the *RAMA* locus precluded facile floxing of the entire coding sequence in a single manipulation as previously performed with other *P. falciparum* genes [4], so we took an alternative approach in which we inserted into exon 2 a synthetic construct combining two heterologous *loxPint* elements (i.e. synthetic introns each containing a *loxP* site; [45]), flanking 176 bp of intervening recodonised exon sequence. The inserted sequence was precisely integrated in a marker-free manner in a single Cas9-promoted homologous recombination step (Fig 1A). The resulting modified *RAMA* gene (referred to as *RAMAloxP*) thus effectively comprised 7 exons with an internal recodonised exon flanked by *loxPint* sequences. DiCre-mediated recombination between the introduced *loxP* sites was predicted to excise this exon. Importantly, excision was also expected to create a frame-shift in the coding sequence immediately downstream of the single chimeric *loxPint* site remaining after excision, leading to a severely truncated *RAMA* coding sequence encoding only the N-terminal 220 residues, referred to below as RAMAΔE2.

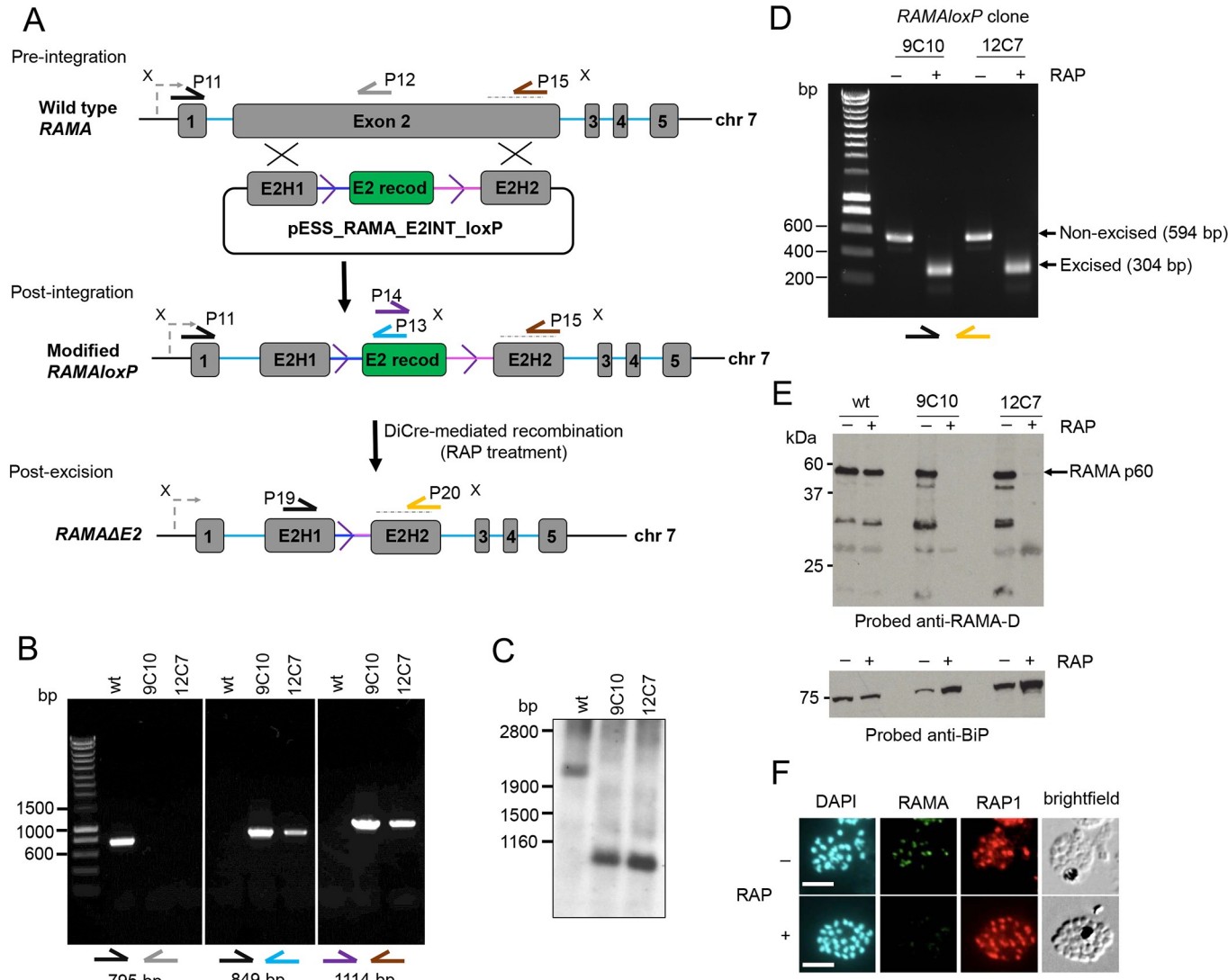

**Fig 1. Conditional disruption of the *P. falciparum RAMA* gene.** (A) The *RAMA* gene (PlasmoDB PF3D7_0707300) comprises 5 exons (numbered grey boxes; not to scale) and 4 introns (blue lines). Two AG-AT motifs (splice site junction consensus sequences; [72] in exon 2, separated by 176 bp, were chosen as the location for insertion of two *loxP*-containing (purple arrowhead) heterologous introns (*SERA2:loxPint*, blue line; *SUB2:loxPint*, pink line) flanking a recodonised form of the intervening 176 bp coding sequence (E2 recod, green box). The integration sequence, provided on repair construct pESS_RAMA_E2INT_loxP, was inserted by double crossover homologous recombination between homology arms (E2H1 and E2H2), facilitated using Cas9 and sgRNAs encoded on a separate plasmid to generate a targeted double stranded break (see Materials and methods). DiCre-mediated recombination between the *loxP* sites in the modified *RAMAloxP* locus was expected to excise the E2 recod sequence, leaving only a single chimeric *SERA2:loxP:SUB2* intron on the chromosome and generating a frame-shift following the first 220 amino acid codons of the *RAMA* coding sequence. Positions of primers used for diagnostic PCR are indicated (half arrows; see S1 Table for numerical codes and sequences of all primers used in this study), as are XmnI restriction sites (X) used for Southern blot analysis. The grey dotted line indicates the region used for hybridisation of a probe for Southern blot. (B) PCR analysis of *RAMAloxP* clones 9C10 and 12C7 confirmed the expected gene modification events. Amplicons diagnostic of the wild type locus obtained from parental 1G5DC parasites (wild type; wt) were not amplified from the *RAMAloxP* clones. Primer identities are colour-coded as in panel (A). (C) Southern blot; XmnI-digested genomic DNA from *RAMAloxP* clones 9C10 and 12C7 was hybridised with a probe specific for the E2H2 sequence. Expected fragment sizes are 2,191 bp (wt) and 1,050 bp (*RAMAloxP*). (D) Diagnostic PCR showing efficient RAP-induced excision of the floxed sequence in the *RAMAloxP* parasite clones, leading in both cases to the expected decrease in amplicon size from 594 bp to 304 bp. Primer identities are colour-coded as in panel (A). (E) Immunoblots of SDS PAGE-fractionated cycle 0 schizont extracts probed with anti-RAMA-D antibodies (raised against residues 482–758 of the RAMA sequence; [36]). Loss of RAMA p60 expression (arrowed) was almost complete in the RAP-treated *RAMAloxP* parasites. Similarly-treated 1G5DC schizonts (wt) were analysed in parallel to confirm that RAP-treatment has no effect on RAMA expression in the absence of the *RAMAloxP* modification. The blots were stripped and re-probed with anti-BiP antibodies as a loading control (lower panels). (F) Representative IFA images demonstrating loss of RAMA expression upon RAP-treatment of *RAMAloxP*-9C10 parasites. Cycle 0 schizonts were co-stained with anti-RAMA-D antibodies and the anti-RAP1 monoclonal antibody (mAb) 4F3. 4,6-diamidino-2-phenylindole (DAPI) was used as a DNA stain. The great majority of RAP-treated parasites (>99.8% as calculated from exhaustive examination of 1,002 schizonts by IFA) showed no detectable anti-RAMA-D staining, but the RAP1 signal remained unaffected. Scale bars, 5 μm.

Limiting dilution cloning of the transfected parasite population resulted in the isolation of 2 parasite clones called *RAMAloxP*-9C10 and *RAMAloxP*-12C7, each independently generated using different sgRNAs. The expected modification of the native *RAMA* locus was confirmed in both parasite clones by diagnostic PCR (Fig 1B) and Southern blot (Fig 1C) and the clones were therefore used for all subsequent experiments. Growth assays comparing replication rates of *RAMAloxP*-9C10 and *RAMAloxP*-12C7 parasites with the parental 1G5DC line revealed no significant differences, indicating that the modifications made to the *RAMA* gene did not detectably impair parasite growth (S1 Fig).

To examine the efficiency of conditional excision of the floxed *RAMA* sequence, tightly synchronised newly-invaded ring stage cultures of both the *RAMAloxP* clones were divided into two and treated for just 4 h with either rapamycin (RAP; 100 nM final) or DMSO vehicle control (1% v/v, mock-treated). Following washing and further incubation for ~44 h to allow maturation to mature schizont stage, parasite genomic DNA (gDNA) was extracted and examined by analytical PCR. This revealed highly efficient RAP-induced excision of the floxed *RAMA* sequence (Fig 1D). To determine the impact of excision on RAMA expression, the schizonts were analysed using a rabbit polyclonal antibody called anti-RAMA-D, specific for a C-terminal segment (amino acid residues 482 to 758) of RAMA [36]. As shown in Fig 1E and Fig 1F, little or no signal was detected by the anti-RAMA-D antibodies in schizonts of the RAP-treated cultures, either by Western blot or by indirect immunofluorescence analysis (IFA). These results confirmed the PCR data, demonstrating in two independent *RAMAloxP* parasite clones essentially complete conditional disruption of RAMA expression within a single erythrocytic cycle.

## RAMA is required for *P. falciparum* replication *in vitro*

The IFA images in Fig 1F indicated that, despite efficient disruption of the *RAMA* gene in RAP-treated *RAMAloxP* parasites, the parasites were able to form multinucleated schizonts towards the end of the erythrocytic growth cycle in which they were RAP-treated (henceforth referred to as cycle 0). To examine the effects of gene disruption on longer-term parasite viability, the mock- and RAP-treated *RAMAloxP* clones were maintained in culture and parasite replication monitored by flow cytometry over the ensuing erythrocytic cycles. Mock- and RAP-treated parental 1G5DC parasites were also included in these experiments to control for any off-target effects of RAP treatment. As shown in Fig 2A, a dramatic reduction in parasite replication rate was quickly evident in the RAP-treated *RAMAloxP* parasites, indicating a defect in parasite proliferation. This was confirmed by diagnostic PCR (Fig 2B) which revealed that upon passage of the cultures for up 9 erythrocytic cycles, the excised locus quickly became undetectable in RAP-treated cultures (indicating disappearance of the RAMAΔE2 mutants) whilst in contrast the initially very minor fraction of non-excised parasites gradually expanded to take over the cultures. This was consistent with an acute fitness defect in the RAMAΔE2 mutants. A further quantitative assessment of the impact of *RAMA* disruption upon parasite viability was obtained by comparing the capacity of mock- and RAP-treated *RAMAloxP* parasites to form zones of erythrocyte lysis, or plaques [46], in static cultures in flat-bottomed microplates (Fig 2C). This showed that RAP-treatment of both the *RAMAloxP*-9C10 and *RAMAloxP*-12C7 clones produced a highly significant reduction in plaque formation; the mean relative plaque forming capacity (RPFC) of RAP-treated parasites was 3.39% ± 0.833 (meaning that for every 10,000 plaques produced by the control mock-treated cultures, only 339 were formed in the same number of RAP-treated culture-containing wells) (p = 3.32 x $10^{-8}$, t = 115.9, d.f. = 4). Together, these data confirmed that RAMA plays a critical role in *P. falciparum* asexual blood stage viability.

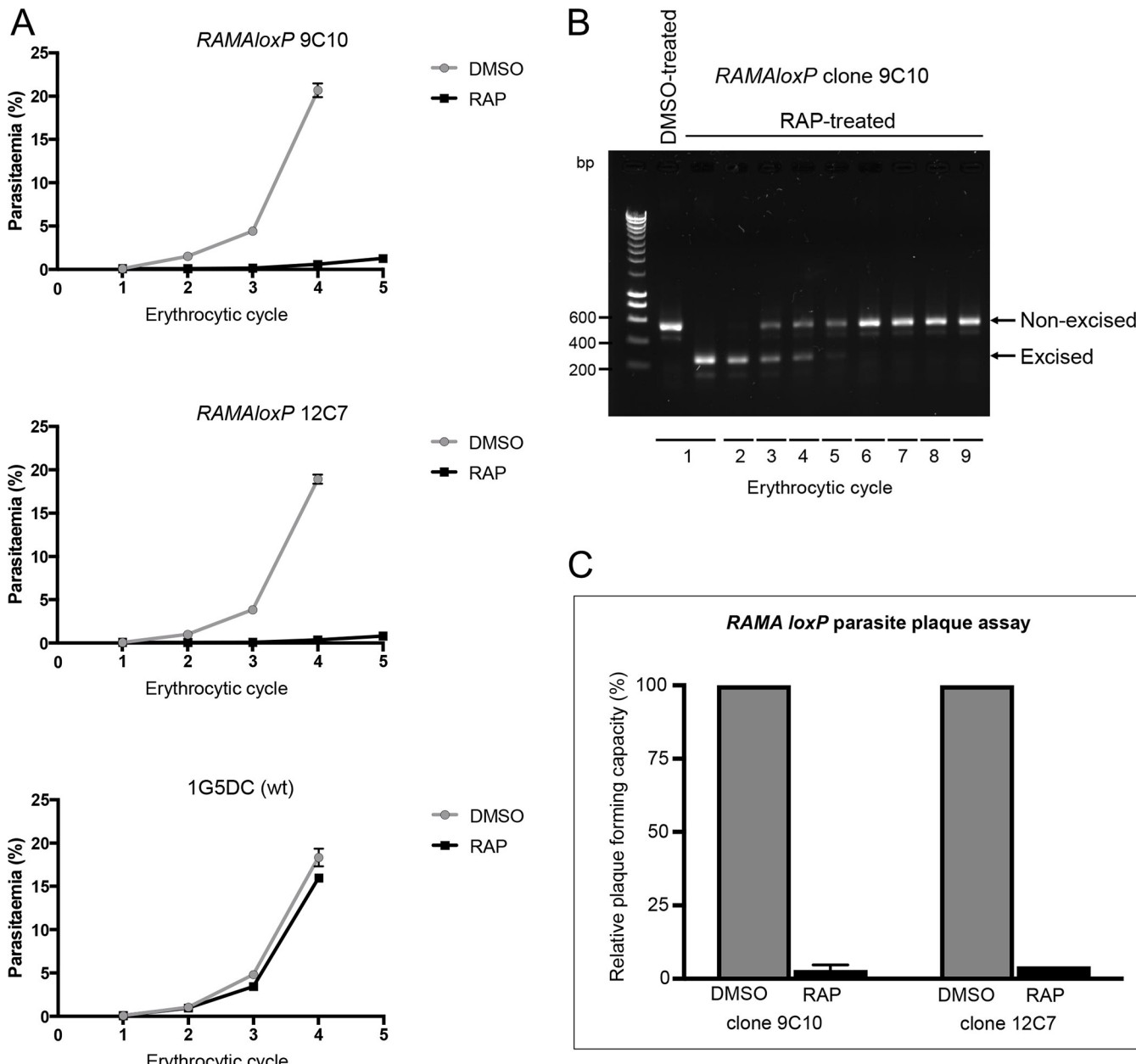

**Fig 2. RAMA is required for asexual blood stage *P. falciparum* replication.** (A) Replication of *RAMAloxP* and parental 1G5DC parasites over the course of 4–5 erythrocytic cycles following mock-treatment (DMSO) or RAP-treatment of synchronous ring-stage cultures. Parasitaemia values (quantified by flow cytometry) were averaged from three biological replicate experiments performed using blood from different donors, and are presented as mean ± SD. Post-hoc comparisons by three-way ANOVA revealed significant differences between growth curves of the mock- and RAP-treated *RAMAloxP* clones 9C10 (p = 0.00995) and 12C7 (p = 0.00860), but not of the parental 1G5DC line (p = 0.998). (B) Diagnostic PCR analysis of mock- or RAP-treated cultures of *RAMAloxP* clone 9C10 parasites, indicating gradual overgrowth of the excised population by an initially undetectable minority of non-excised parasites. Parasite gDNA sampled at each cycle over the course of 9 erythrocytic growth cycles was analysed (see Fig 1A for primer identities). The signal specific for the excised locus became less intense with time whilst the signal specific for the non-excised locus increased in intensity, eventually becoming the sole PCR amplicon detectable. (C) Plots comparing plaque formation by mock- and RAP-treated cultures of *RAMAloxP* clones 9C10 and 12C7. Cultures were dispensed into flat-bottomed microwell plates at equivalent parasite densities and plaques scored following static incubation for 14 days. Error bars represent ± 1 SD from two independent experiments in blood from different donors. A total of 180 wells were used for each treatment in each replicate of the assay.

## Disruption of RAMA produces an invasion defect

Given the established role of rhoptries in invasion, we considered that the replication defect shown by the RAMAΔE2 parasites might most likely be explained by a reduced capacity to invade host cells. To investigate this, mature *RAMAloxP*-9C10 and *RAMAloxP*-12C7 cycle 0 schizonts were isolated from highly synchronous mock- or RAP-treated cultures then added to fresh RBCs to achieve a similar starting parasitaemia. Following further incubation for 4 h to allow schizont rupture, the cultures were stained with Hoechst 33342 and the proportions of newly-invaded ring-stage parasites determined by flow cytometry. This consistently showed that the RAP-treated parasites displayed substantially reduced ring formation, corresponding to only ~10% of that of mock-treated parasites (Fig 3A).

As egress and invasion are intimately linked, we explored whether the lack of ring formation in these assays was indicative of a defect in egress from the host erythrocyte. To do this the capacity of RAMAΔE2 parasites to undergo egress was examined by time-lapse differential interference contrast (DIC) microscopy. As shown in Fig 3B, no detectable differences in the efficiency, kinetics or morphology of merozoite egress from RAP-treated *RAMAloxP* parasites was evident compared to their mock-treated counterparts. It was concluded that disruption of RAMA does not affect egress, but results in a severe defect in productive invasion of new host erythrocytes.

To characterize this invasion defect in further detail, the behaviour of naturally-released merozoites and their interactions with host RBCs was examined by time-lapse DIC video microscopy. As shown in S1 Movie and S2 Movie, and quantified in Fig 3C, merozoites released from RAP-treated (RAMAΔE2) cycle 0 schizonts interacted initially with neighbouring host RBCs in a similar manner to their mock-treated *RAMAloxP* counterparts, inducing repeated and often strong deformation of the host cells. However, unlike the mock-treated merozoites, in no case (out of a total of 20 similar egress events examined) did we observe successful invasion by the RAMAΔE2 merozoites. Of particular additional interest, no RBC echinocytosis was ever observed in the case of the RAMAΔE2 merozoite-RBC interactions. Echinocytosis generally takes place shortly following invasion (see S1 Movie), but under certain conditions can also occur in the absence of invasion, where it is thought to indicate discharge of rhoptry components [3–5]. It was concluded that loss of RAMA selectively results in a fatal block in invasion, likely due to a defect in rhoptry discharge.

## Disruption of RAMA leads to mislocalisation of some but not all rhoptry neck and rhoptry bulb proteins

As mentioned above, morphological characterisation of mature cycle 0 RAMAΔE2 schizonts by light microscopy (IFA) indicated apparently normal schizont development (Fig 1F). However, given the clear invasion phenotype displayed by the RAMAΔE2 mutants and the known rhoptry bulb localisation of RAMA [36], we decided to examine the structure of the mutant parasites in greater detail to assess the possible impact of RAMA disruption on the makeup and morphology of the rhoptries. This was considered particularly important given previous claims that RAMA interacts with RhopH3 and RAP1 and plays a role in the trafficking of both proteins to the rhoptries [36, 37].

To determine the effects of RAMA disruption on the expression and trafficking of other rhoptry proteins, mature cycle 0 schizonts from mock-treated and RAP-treated *RAMAloxP* cultures were probed by IFA using a suite of antibodies specific for a range of rhoptry bulb proteins. As shown in Fig 4A, this revealed that the subcellular localisation and staining intensity of the LMW rhoptry complex protein RAP2, the HMW rhoptry complex proteins RhopH1/Clag3.1, RhopH2 and RhopH3, the invasion ligand Rh5, and the rhoptry-associated

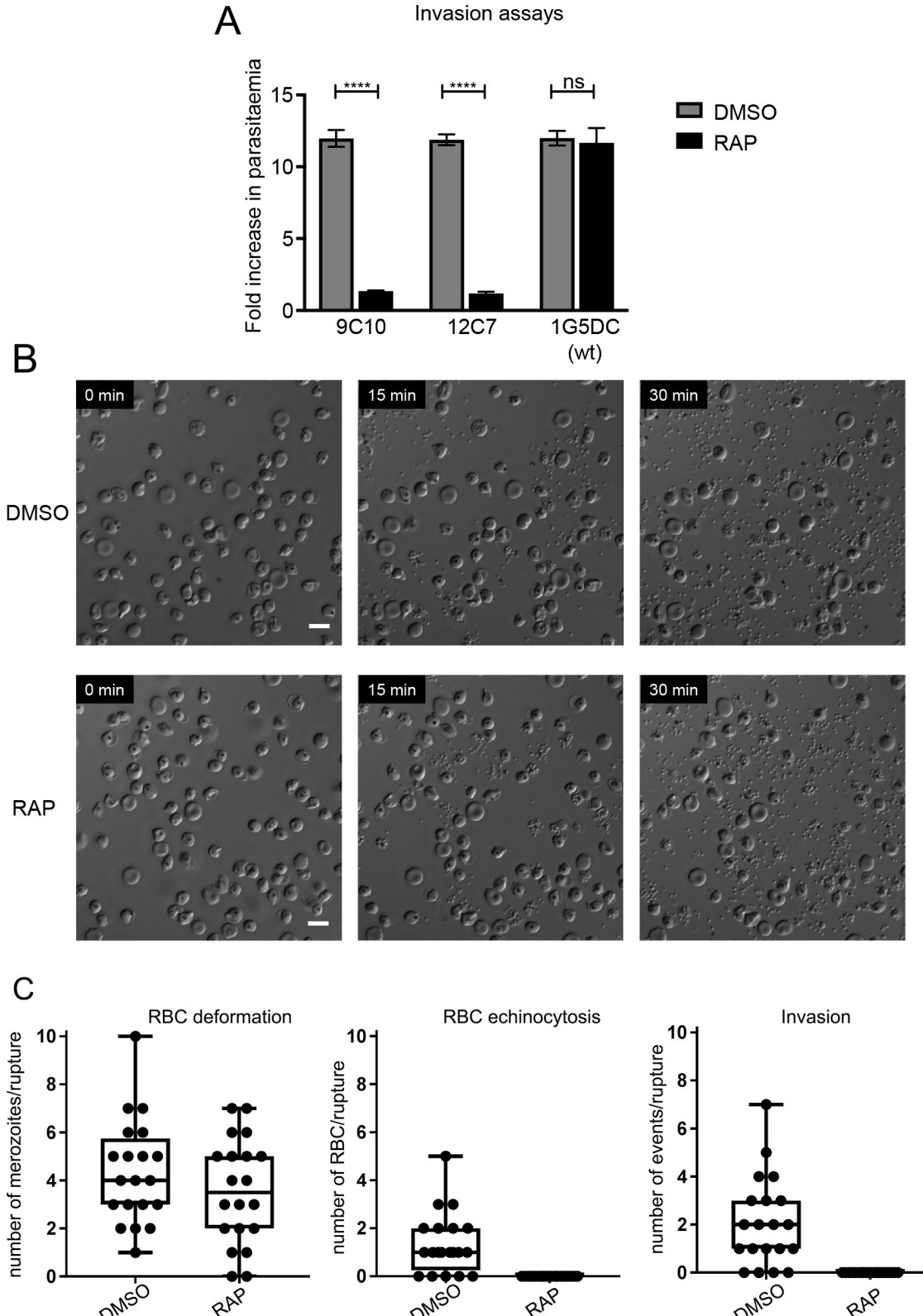

**Fig 3. Disruption of RAMA results in an invasion defect.** (A) Reduced ring formation by RAMA mutants. Mature cycle 0 schizonts isolated from mock- or RAP-treated cultures of the indicated clones were incubated with fresh RBCs and ring-stage parasitaemia determined 4 h later by flow cytometry. Data were averaged from three biological replicate experiments, using blood

from different donors. Error bars, ± 1 SD. Statistical significance was determined by a two-tailed unpaired t-test where p≤0.0001, **** and p>0.05, non-significant (ns). Ring formation by the parental 1G5DC parasites was unaffected by RAP treatment (t = 0.507, d.f. = 4, p = 0.639). (B) Still images from experiments monitoring egress of merozoites by time-lapse DIC microscopy of mature mock- and RAP-treated *RAMAloxP* clone 9C10 schizonts. Highly synchronous cycle 0 schizonts were incubated with the parasite cGMP-dependent protein kinase G inhibitor (4-[7-[(dimethylamino)methyl]-2-(4-fluorphenyl)imidazo[1,2-α]pyridine-3-yl]pyrimidin-2-amine (compound 2; C2) for 4 h to reversibly arrest egress [73]. Immediately after washing away the C2 the parasites were monitored for 30 minutes. No delay in egress was observed in the RAP-treated population, and there were no discernible differences in egress efficiency or morphology. Time in minutes from commencement of microscopy is indicated. Scale bar, 10 μm. Images shown are from clone 9C10 but are representative of 3 independent replicate experiments examining *RAMAloxP* clones 9C10 and 12C7. (C) Quantification of invasion, merozoite-induced RBC surface deformation and echinocytosis observed by video microscopy following rupture of mock- or RAP-treated *RAMAloxP* clone 9C10 schizonts. At least 20 videos per condition were quantified. Statistical significance was assessed by t-test; ns indicates not significant (p > 0.05) whereas **** indicates p < 0.0001. The RAP-treated (RAMAΔE2) merozoites interacted with and deformed target RBCs, but neither invaded them nor induced echinocytosis. See also S1 Movie and S2 Movie.

protein ARO (thought to localise to the cytosolic face of the rhoptry bulb and neck [47–49]) were unaffected by RAMA disruption; all retained their characteristic apical punctate rhoptry signal in the RAP-treated *RAMAloxP* schizonts. Note that similar findings were noted above for RAP1 (Fig 1F). It was concluded that truncation of RAMA does not discernibly affect the trafficking of a number of established rhoptry bulb proteins.

To examine the effects of RAMA truncation upon the subcellular localisation of RON proteins, mock- and RAP-treated *RAMAloxP* schizonts were next probed with antibodies specific for RON2 [50], RON3, or RON4 [51]. As shown in Fig 4B and Fig 4C, in all cases this revealed evidence for RON protein mislocalisation in the RAMAΔE2 mutants, although the precise phenotype appeared to depend upon the extent of parasite maturity. Specifically, for all three RON proteins, in immature schizonts a much reduced signal intensity was detectable which lacked the punctate, apically-disposed pattern typical of rhoptries. In highly mature segmented schizonts, IFA signals for RON2, RON3 and RON4 were often completely absent. These findings were particularly intriguing since, whilst RON2 and RON4 are established rhoptry neck proteins that play roles in TJ formation, RON3 has been localised by immuno electron microscopy to the rhoptry bulb in *Plasmodium* [52], rather than to the rhoptry neck as originally determined for the *Toxoplasma* orthologue [11]. Work in *Toxoplasma* also supports a rhoptry bulb location (see ToxoDB annotation for the *TgRON3* gene at https://toxodb.org/toxo/app/record/gene/TGME49_223920). The current nomenclature for RON3 is therefore somewhat misleading, and these results therefore suggested that disruption of RAMA expression can modulate the localisation of rhoptry neck proteins as well as at least one rhoptry bulb protein.

To investigate this issue further using additional rhoptry markers, we examined *RAMAloxP* schizonts by IFA using antibodies to two proteins that have been previously localised to either the rhoptry neck (RON12; [21]) or both the rhoptry body and the neck (Rh2b; [18]) in *P. falciparum* merozoites. As shown in S2 Fig and in contrast to the effects on RON2, RON3 and RON4, disruption of RAMA expression had no discernible effect on expression or localisation of either RON12 or Rh2b.

As expected, disruption of RAMA did not impact on the localisation of the major merozoite surface protein MSP1 or the microneme protein apical membrane antigen 1 (AMA1) (Fig 4D). AMA1 is often observed to translocate onto the merozoite surface in highly mature schizonts, and this phenomenon too was observed at similar frequency in both mock-treated and RAP-treated *RAMAloxP* schizonts (Fig 4D, lower panels). Collectively, these results indicated that disruption of RAMA results in selective mislocalisation and/or loss of rhoptry neck and certain rhoptry bulb proteins, whilst having no detectable impact on the expression and trafficking of other rhoptry proteins examined.

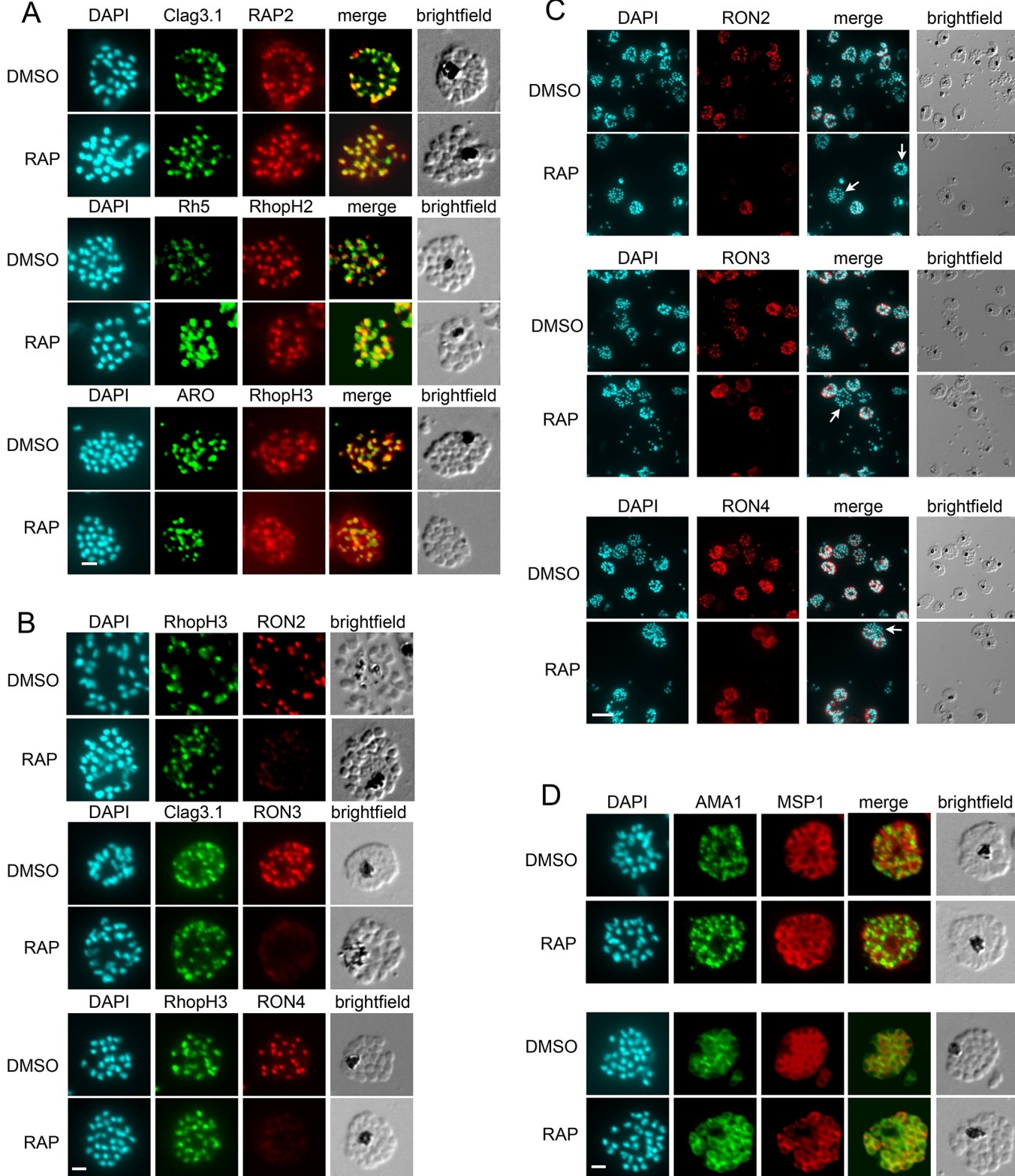

**Fig 4. Disruption of RAMA leads to selective mislocalisation of some but not all rhoptry proteins.** (A) IFA showing that the staining profile of RAP2, RhopH1/Clag3.1, Rh5, RhopH2, ARO and RhopH3 was unaltered between DMSO- and RAP-treated *RAMAloxP* clone 9C10 schizonts. Scale bar, 2 μm. (B) IFA showing that, whilst RhopH3 and RhopH1/Clag3.1 staining was unaffected by RAMA disruption, RON2, RON3 and RON4 staining was modified in RAP-treated *RAMAloxP* clone 9C10 schizonts. In all cases, the staining was much weaker in the RAMAΔE2 parasites and lacked the characteristically punctate,

apically-disposed pattern typical of rhoptry staining. Scale bar, 2 μm. (C) The extent of RON protein mis-trafficking depends on the maturity of the RAP-treated *RAMAloxP* schizonts. Whilst some signal was still detectable in less mature schizonts (those with fewer nuclei), the RON2, RON3 and RON4-specific signal was absent or much reduced in the mature segmented schizonts (examples indicated with arrows). Note that these images are of lower magnification that those shown in the rest of the figure in order to display multiple schizonts in each field. Scale bar, 10 μm. (D) Schizonts of DMSO- and RAP-treated *RAMAloxP* clone 9C10 were probed with a polyclonal antibody against AMA1 as well as anti-MSP1 antibodies. AMA1 localisation (upper panel) and discharge onto the merozoite surface in mature schizonts (lower panel) was unaffected by RAMA disruption. Scale bar, 2 μm. Parasite nuclei were visualised throughout by staining with DAPI.

## RAMA is selectively required for formation of the rhoptry neck

Previous studies of RAMA suggested a role in rhoptry biogenesis [36]. Our discovery that some rhoptry proteins become mislocalised upon RAMA disruption prompted us to next examine the impact of RAMA disruption on rhoptry formation and morphology. To do this, mature segmented cycle 0 mock- or RAP-treated *RAMAloxP* schizonts were examined by TEM. To quantify rhoptry formation, the number of rhoptry profiles detectable in individual intracellular merozoites in 70 nm thin sections was recorded. This revealed no statistically significant differences in the total number of visible rhoptry profiles between the control and RAMAΔE2 parasites (glm fit with Poisson model, predicting rhoptry count from status, non-zero status coefficient test p-value = 0.232) (Fig 5A). This result was in complete accord with the IFA results obtained with the rhoptry bulb-specific antibodies described in Fig 4, suggesting that the overall number of rhoptry organelles assembled in wild-type and RAMAΔE2 parasites was similar.

During the above analysis, we noticed that whilst other intracellular features of the control and RAP-treated parasites (including microneme morphology) were similar, there was a relative absence of typical club-shaped rhoptry profiles in the RAMAΔE2 TEM micrographs. To attempt to quantify these apparent differences in rhoptry morphology, the TEM rhoptry images were categorised as being either circular or club-shaped based on blinded visual examination (Fig 5B). As shown in Fig 5C, whilst rhoptries of both shapes were observed in merozoites of mock-treated and RAP-treated *RAMAloxP* parasites, a significantly lower proportion of club-shaped profiles was observed in the RAMAΔE2 samples. Moreover, in those RAMAΔE2 rhoptries that visibly possessed an apical projection, these appeared qualitatively different from wild type, being relatively short and stubby compared to the elongated projections typical of the latter (Fig 5B). This difference in shape was confirmed by quantifying the aspect ratio (the ratio of the longest to shortest diameter) in the electron micrographs of all non-circular rhoptries from both control and RAP-treated parasites. The resulting data (Fig 5D) clearly indicated that the length of the rhoptry neck projections in control parasites (mean aspect ratio = 1.73) was significantly greater than those of RAP-treated parasites (mean aspect ratio = 1.21) (p <0.0001, t = 5.29, d.f. = 58). Since these profiles represent random section planes through rhoptries, it was considered unlikely that the differences noted were a chance result of random variation in the section plane obtained during sample preparation. However, to finally examine this point, electron tomography was performed on serial 200 nm thick sections of a single mock-treated *RAMAloxP* schizont and a RAP-treated *RAMAloxP* schizont (S3 Movie and S4 Movie). This allowed computational three-dimensional modelling of much of the rhoptry content of each schizont by segmentation and rendering of the tomograms. The resulting models (Fig 5E) supported the thin section TEM data in showing a clear difference between the elongated shape of rhoptries in the DMSO-treated *RAMAloxP* schizonts and the strikingly more spherical nature of the RAMAΔE2 rhoptries.

Collectively, our EM data indicated that disruption of RAMA does not affect the number of rhoptries formed, but has a major impact on their morphology, leading to the generation of

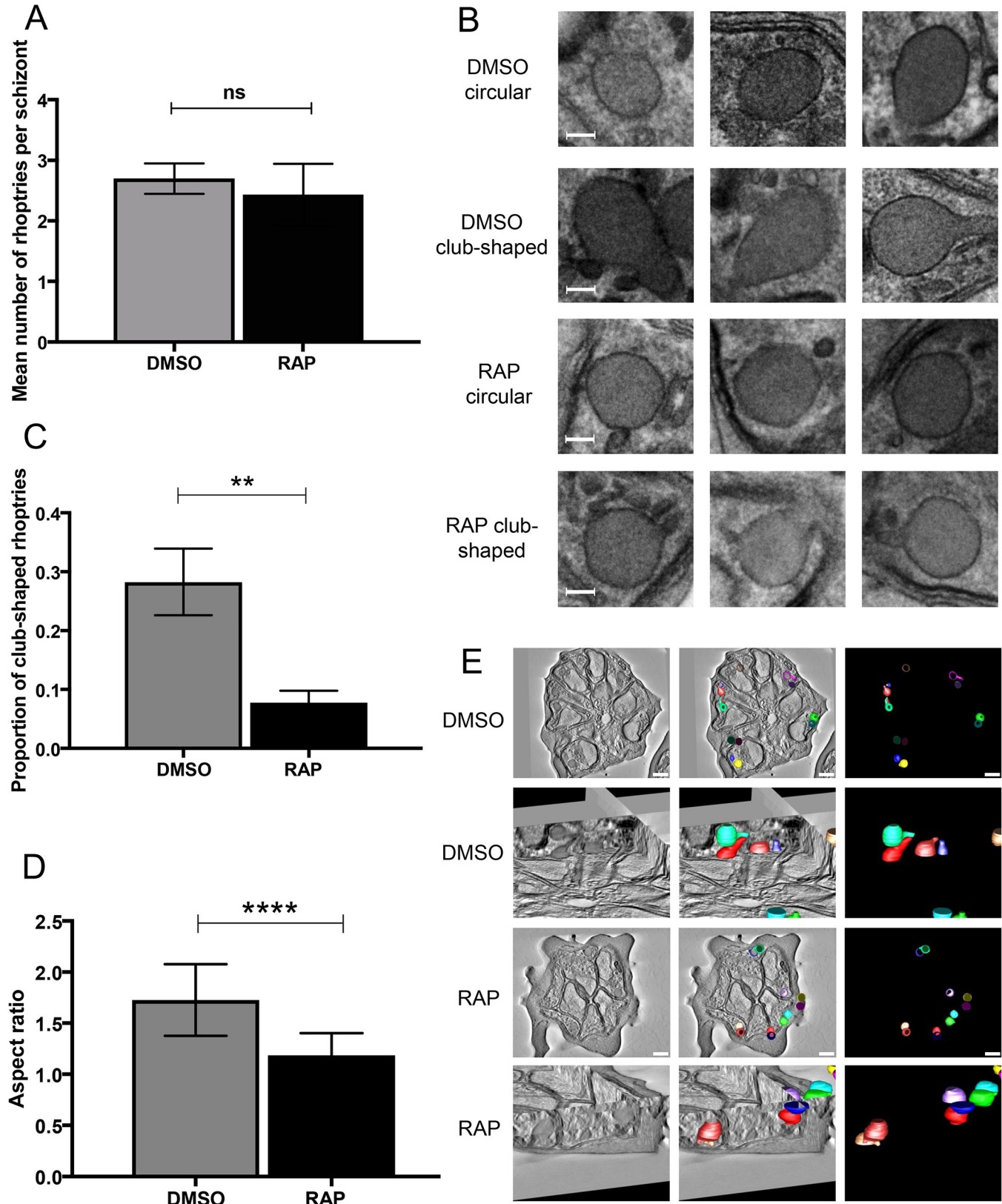

**Fig 5. Disruption of RAMA leads to defective rhoptry morphology and reduced rhoptry neck formation.** Tightly synchronised mature schizonts from DMSO- or RAP-treated cultures of *RAMAloxP* clone 9C10 parasites were fixed and processed for TEM, then three sections from each sample taken and 20

schizonts examined per section. (A) Mean number of visible rhoptry profiles per schizont calculated from each set of 60 TEM images of DMSO-treated and RAP-treated schizonts. There was no statistically significant difference in the mean number of rhoptry profiles visible. Error bars, ±1 SD. p>0.05, non-significant (ns). (B) Representative TEM images from DMSO and RAP-treated *RAMAloxP* schizonts indicating that club-shaped and circular (i.e. likely spherical) rhoptry profiles were detected in both. Scale bar, 100 nm. (C) The proportion of club-shaped rhoptry profiles visible in DMSO-treated *RAMAloxP* schizonts was greater than that in RAP-treated *RAMAloxP* schizonts. Statistical significance was determined by unpaired t-test. Error bars depict ±1 SD. p≤0.01, **. (D) Plots of aspect ratio (longest diameter divided by shortest diameter) calculated from thin section electron micrographs for all rhoptries (both DMSO and RAP-treated) classified as non-circular. Non-circular rhoptries from DMSO-treated schizonts displayed a higher aspect ratio (i.e. were more elongated) than those from RAP-treated *RAMAloxP* schizonts. Statistical difference determined by two-tailed t-test. Error bars, ±1SD. p≤0.0001, ****. (E) Two dimensional cross section images and three-dimensional models derived from 200 nm thick section serial tomograms acquired from a DMSO-treated (upper two panels) and a RAP-treated *RAMAloxP* schizont (lower two panels). Rhoptries are depicted in various colours to easily distinguish individual structures. Whilst the rhoptries of the DMSO-treated schizonts appear predominantly club shaped, those of the RAP-treated parasites appear much more circular with stubby, non-conoid projections. Scale bars, 500 nm. All observations where zero rhoptry profiles were reported for a merozoite were deemed 'failed' observations for the purposes of this analysis and were not included in the analysis.

relatively spherical or 'neckless' rhoptries. This may explain why certain rhoptry neck proteins mislocalise in the RAMA mutants.

## Discussion

The involvement of rhoptry proteins in host cell entry has long been proposed, supported by early observations that rhoptry discharge coincides temporally with invasion in several api-complexan parasites, including *Toxoplasma* [53–56] and *P. falciparum* [17]. Those studies have been burgeoned by recent demonstrations that correct positioning of rhoptries at the api-cal pole of the *Toxoplasma* tachyzoite is a prerequisite for invasion (but not egress) [48] whilst in both *P. falciparum* merozoites and *Toxoplasma* tachyzoites rhoptries appear to undergo fusion with each other during invasion [30, 57]. The discovery of the role of several RON pro-teins in TJ formation [15, 16] was consistent with this overall model for rhoptry function, but as discussed in the Introduction it is also now clear that rhoptry proteins play roles following completion of invasion in such diverse functions as PVM generation, subversion of host cell signalling, and nutrient acquisition. Here we have established a new role for a *Plasmodium* rhoptry protein in biogenesis of these important organelles.

We draw three major conclusions from our study. First, we have shown that RAMA is essential for parasite survival, and that disruption of RAMA expression results in a selective, fatal defect in host cell invasion with no obvious effect on intracellular parasite replication or egress. This finding is reminiscent of the results of conditional ablation in *Toxoplasma* of ARO, a palmitoylated protein localised to the cytoplasmic rhoptry surface which is essential for correct tethering of the rhoptries within the tachyzoite apex [47–49]. Loss of TgARO results in rhoptries that are dispersed throughout the cytoplasm of the parasite, and the mutant tachy-zoites undergo normal egress but cannot invade host cells. As well as not being able to pene-trate targeted RBCs, we found that RAMAΔE2 merozoites showed no capacity to induce echinocytosis in interacting RBCs. This provides important insights into the underlying defect. Evidence that rhoptry discharge can take place upon contact with the host cell even in the absence of productive invasion was first demonstrated through observations of the appearance of small nascent intraerythrocytic channels or vacuoles at the site of apical attachment of *P. knowlesi* merozoites to the RBC surface; this occurred both under normal, invasion-permissive conditions but also in cultures containing cytochalasin B, an inhibitor of actin polymerisation that blocks invasion downstream of TJ formation [58–60]. Subsequent work in *P. falciparum* combining electron microscopy with super-resolution immunofluorescence microscopy showed that these vacuoles contain rhoptry proteins [13]. Similar structures (termed 'eva-cuoles' when formed in the presence of cytochalasin) were observed upon interaction of *Toxo-plasma* tachyzoites with host cells, and these were also found to contain large amounts of rhoptry bulb proteins [53, 61]. More recent comprehensive microscopic examination of the

interactions between free *P. falciparum* merozoites and host RBCs provided evidence that rhoptry discharge is the primary inducer of echinocytosis in target RBCs [3], and consistent with the above observations this can occur even in the absence of productive invasion. Very recent work has added further support to this model, showing that merozoites genetically deficient in key components of the cyclic nucleotide signalling pathway or the molecular motor that drives invasion can induce echinocytosis in target RBCs, despite being unable to penetrate the cells [4, 5]. It has thus become clear that host RBC echinocytosis—irrespective of whether it is followed by successful invasion—provides a useful reporter for rhoptry function in *Plasmodium*. In our current study, the complete absence of echinocytosis upon interactions between RAMAΔE2 merozoites and target RBCs is therefore completely consistent with a defect in rhoptry discharge and/or the absence of the (currently unidentified) rhoptry component(s) that induce echinocytosis. RAMA was previously proposed to act as an invasion ligand based on indications that anti-RAMA antibodies inhibit merozoite invasion of RBCs [36] and that recombinant RAMA-based peptides could apparently bind to RBCs and inhibit invasion [62]. Our study did not directly address this proposed function for RAMA.

The second clear conclusion from our study is that RAMA is not required for correct trafficking of RAP1 or RhopH3, as previously proposed based on protein-protein interaction and co-localisation studies [36–38], nor for trafficking of several other rhoptry bulb proteins examined. Our results are therefore not consistent with a rhoptry bulb-specific protein escorter role for RAMA. However, our findings may not necessarily be in conflict with the interaction model; RAMA may associate with RhopH3 and RAP1, for example, in order to facilitate formation of the respective HMW and LMW complexes, whilst other factors may regulate localisation of these protein complexes to the rhoptry bulb. It is important to note that our RAMA disruption strategy did not remove the entire gene and could in principle result in the expression of the N-terminal 220 residues of the protein as a truncated protein product (Fig 1). We did not have the antibody tools to detect whether such a truncated product was expressed in our mutants. However, were it to be expressed, the truncated protein would lack the C-terminal sequences between residues 315–840 previously shown to interact with RAP1 and sortilin, and required for trafficking to the rhoptries [36–38]. We are therefore confident that our strategy generated a loss-of-function mutant, and the phenotype supports that.

Third, our results show that RAMA is instead required for correct localisation of a subset of RON and rhoptry bulb proteins and for formation of the rhoptry neck, providing the first genetic evidence that RAMA plays an essential role in rhoptry biogenesis. The causal relationship between these phenotypes is unclear, in that we cannot discriminate between whether a defective rhoptry morphology leads to mislocalisation of the RON proteins or alternatively whether correct trafficking of RON proteins is required for rhoptry neck morphogenesis. RAMA may interact with RON proteins to ensure their correct trafficking, although no associations between RAMA and RON proteins were identified in the earlier studies [36, 37]. The involvement of a rhoptry bulb protein in trafficking of rhoptry neck proteins appears somewhat counterintuitive. Could RAMA direct these proteins to the bulb then regulate their onward trafficking to the nascent neck domain? It is interesting to note that RON12, a rhoptry neck protein localisation of which was unaffected by RAMA disruption in this study, has very recently been confirmed as a rhoptry neck protein in merozoites of the rodent malaria parasite *Plasmodium berghei*, but a rhoptry bulb protein in *P. berghei* sporozoites, a developmental stage of the parasite generated in the mosquito vector [63]. This suggests that the trafficking characteristics of rhoptry proteins may not be an intrinsic function of their structure *per se*, but dependent upon the developmental context of their expression.

Both RON2 and RON4 are key for formation of the TJ, so their mis-trafficking may explain in part the invasion defect in the RAMAΔE2 mutants. However, we propose that the most

dramatic element of the phenotype–the defect in rhoptry neck formation–is most likely primarily responsible for the loss of invasive capacity in the mutants. Absence of the neck structure would presumably prevent docking and fusion of the rhoptries with the parasite plasma membrane at the merozoite apical prominence, reducing or ablating the capacity for rhoptry discharge. It might be expected that loss of the rhoptry neck structure would also result in a phenotype similar to that of loss of TgARO [48, 49], with the rhoptries becoming dispersed throughout the merozoite cytoplasm rather than apically located; however the relatively small dimensions of the *P. falciparum* merozoite relative to the *Toxoplasma* tachyzoite likely limits cytoplasmic movement of untethered rhoptries, rendering this phenotype less obvious.

In conclusion, the selective impact of RAMA disruption on the correct localisation of some but not all rhoptry proteins supports the existence of subsets of rhoptry proteins, some of which require RAMA for correct trafficking whilst others do not. We have shown that RAMA disruption is lethal, likely through a direct impact upon rhoptry biogenesis and a resulting indirect impact upon invasion but not egress. Given that there is no obvious orthologue of RAMA in *Toxoplasma* and other apicomplexan parasites outside of the *Plasmodium* genus, our findings raise obvious questions as to how rhoptry neck formation is regulated in other Apicomplexa. In this regard is interesting to note that disruption of the gene encoding a GPI-anchored *Toxoplasma* rhoptry bulb protein called TgCA_RP results in dysmorphic, fragmented rhoptries [64], although the phenotype is quite different from that of the RAMAΔE2 *P. falciparum* mutants and the morphologies of rhoptries are also distinct between the two apicomplexan genera. TgCA_RP and RAMA share no homology, but it is conceivable that the roles of the two proteins in rhoptry biogenesis may be similar.

## Materials and methods

### Ethics statement

This work used human blood that was provided anonymously through the UK Blood Transfusion Service.

### Reagents and antibodies

Oligonucleotide primers (see S1 Table) were purchased from Sigma-Aldrich (Gillingham, UK). Rapamycin was also obtained from Sigma-Aldrich and stored frozen as a stock solution (10 μM) in DMSO. The antifolate drug WR99210 (Jacobus Pharmaceuticals, Princeton, NJ) was stored as a 20 μM stock solution in DMSO. The anti-RAMA-D, anti-RhopH3 and anti-KAHRP antibodies were kind gifts of Ross Coppel (Monash University, Australia) whilst Osamu Kaneko (Nagasaki University, Japan) kindly provided antibodies against RhopH1/Clag3.1. The polyclonal anti-AMA1 antibody has previously been described [65], as has the RhopH2-specific mAb 61.3 [66]. The anti-MSP1 mAb 89.1, the anti-RON3 mAb 1H1, the anti-RAP1 mAb 4F3, a rabbit anti-ARO polyclonal antibody, and rabbit antibodies against RON12 were kind gifts of Tony Holder (Francis Crick Institute, UK), and the antibodies to Rh2b were generously provided by Julian Rayner (University of Cambridge, UK). Other antibodies were generously provided by John Vakonakis, University of Oxford, UK (anti-MAHRP1 antibody), Simon Draper, University of Oxford, UK (anti-Rh5 antibody), Takafumi Tsuboi, Ehime University Japan (anti-RON2 antibody) and Jean-François Dubremetz, Université Montpellier 2, France (anti-RON4 mAb 24C6).

## Generation of RAMA conditional mutant parasite lines

*RAMAloxP* mutant *P. falciparum* lines were generated by transfecting parasites with 60 μg of a repair plasmid alongside 20 μg of a plasmid for expression of Cas9. Repair plasmid pESS_RA-MA_E2INT_loxP (synthesised by ThermoFisher Scientific) comprised synthetic heterologous *SERA2:loxP* and *SUB2:loxP* introns which were inserted at AG-AT sites 776 bp and 952 bp respectively downstream of the *RAMA* gene ATG start codon. The intervening 174 bp was codon-optimised for expression in *Escherichia coli*. Sequence comprising 568 bp of exon 2 both 5' and 3' to the site of the heterologous intron insertions was included to act as flanking regions for homologous recombination. EuPaGDT software was used to identify 20 bp proto-spacer sequences (flanked by a 5'-NGG-3' protospacer adjacent motif) in order to specifically target Cas9-mediated cleavage to the *RAMA* gene locus. pRAMAsgRNA1 and pRAMAsgRN A2 were created by modifying the Cas9-sgRNA-encoding pDC2 vector (a kind gift of Marcus Lee, Sanger Institute, UK) by introduction of sgRNAs 1 and 2 generated by annealing comple-mentary primer sequences RAMA_sgRNA_E2_F to RAMA_sgRNA_E2_R (sgRNA1), and RAMA_sgRNA_E2S_F to RAMA_sgRNA_E2S_R (sgRNA2).

## *P. falciparum* cell culture, transfection and cloning

All parasite work described used the 3D7-derived DiCre-expressing 1G5DiCre (1G5DC) *P. falciparum* clone [44]. Parasites were maintained in culture and synchronised using standard procedures [67, 68]. DNA constructs were introduced by electroporation into mature schiz-ont-stage 1G5DC parasites using previously described methods [69]. Approximately 24 h post-transfection the cultures were supplemented with WR99210 (2.5 nM) to allow selection of parasites carrying the human *dhfr* selectable marker encoded by the pRAMAsgRNA plas-mids. Parasites were maintained under drug selection for 2 erythrocytic cycles before being transferred to drug-free culture medium. Following confirmation of integration in the result-ing parasite populations by diagnostic PCR, parasites were cloned by limiting dilution [46]. Conditional excision of floxed genomic sequences in the transgenic parasite clones was induced as previously described [44] by treating tightly synchronized ring stage parasite cul-tures with 100 nM RAP in 1% (v/v) DMSO (or the same concentration of DMSO only as a vehicle control) for 4 h at 37°C.

## Detection of integration and excision events by diagnostic PCR, Southern blotting and immunoblotting

5′ and 3′ integration of pESS_RAMA_E2INT_loxP into the genome was detected by PCR using the integration specific primers RAMA_exon1_F plus RAMA_exon2recod_R, and RAMA_exon2recod_F plus RAMA_exon2_R. No amplification was expected from the wild type specific primer set RAMA_exon1_F plus RAMA_exon2WT_R. Diagnostic PCR was also used to confirm that the integrated DiCre expression cassette remained unmodified in isolated clones, using primers SERA5_DiCre_F and DiCre_integrated. Reversion to the wild type *SERA5* locus by spontaneous excision of the integrated DiCre cassette (which can occur at low levels in the 1G5DC line; [4]) was indicated by the amplification of a 1700 bp amplicon using primers specific to the unmodified *SERA5* locus (SERA5_DiCre_F and SERA5_DiCre_R). To further confirm the genotype of potential *RAMAloxP* integrant clones, a 400 bp digoxigenin (DIG)-labelled probe corresponding to the 5' 400 bp of the region of exon 2 used as the second homology arm was generated by PCR amplification using primers RAMA_SB_homology2_F and RAMA_SB_homology2_R, performed in the presence of DIG-labelled nucleotides pro-vided by the PCR DIG Labelling Mix Plus (Roche). Approximately 5 μg of gDNA from

putative integrant cultures was digested with an excess of XmnI restriction enzyme, then fractionated by agarose gel electrophoresis, transferred overnight to Amersham Hybond N+ nylon membranes (GE Healthcare Life Sciences), and cross-linked with UV light (1200 μjoules/cm$^2$). After pre-hybridisation, hybridisation, washing and blocking steps, hybridised signal was detected by incubation with polyclonal alkaline phosphate-conjugated anti-DIG antibodies (Roche). The membrane was incubated with detection buffer and CDP-star (Roche), before being exposed to X-ray film. RAP-induced recombination between genomic *loxP* sites was detected by PCR analysis of schizont-stage gDNA (harvested 42–44 h following mock- or RAP-treatment) using primers RAMA_Harm1_F and RAMA_Harm2_R. For immunoblot analysis, SDS-PAGE fractionated proteins were transferred to nitrocellulose membranes before being blocked, incubated with primary antibody diluted to the appropriate concentration, washed and then probed with a HRP-conjugated secondary antibody as previously described [65]. After final washes, membranes were incubated with Immobilon Western Chemiluminescent HRP Substrate (Merck) and signals visualised by exposure to X-ray film.

## Indirect immunofluorescence assays

Thin blood films of mock- or RAP-treated Percoll-enriched *RAMAloxP* schizonts were air-dried and stored desiccated at -80˚C. As required, samples were thawed at 37˚C and parasites fixed in 4% (w/v) paraformaldehyde and permeabilised in 0.1% (v/v) Triton X-100. Fixed slides were blocked then probed with the relevant antibody diluted as follows: rabbit anti-RAMA-D, 1:1000; mouse mAb 4F3 (anti-RAP1), 1:100; rabbit anti-RhopH1/Clag3.1, 1:100; mouse mAb MRA876 (anti-RAP2), 1:200; rabbit anti-Rh5, 1:10,000; mouse mAb 61.3 (anti-RhopH2), 1:100; rabbit anti-ARO, 1:500; rabbit anti-RhopH3 (anti-Ag44), 1:2000; rabbit anti-RON2, 1:250; mouse mAb 1H1 (anti-RON3), 1:100; mouse mAb 24C6 (anti-RON4), 1:500; rabbit anti-AMA1, 1:500; mouse mAb 89.1 (anti-MSP1), 1:1000; rabbit anti-RON12, 1:5000; rabbit anti-Rh2b, 1:500. After incubation and washing, slides were probed with the required Alexa Fluor 488 or 594-conjugated secondary antibody diluted 1:10,000. Slides were then mounted in ProLong Gold Antifade Mountant containing DAPI (ThermoFisher Scientific), sealed with Cytoseal-60 (Thermofisher Scientific) and images collected using a Nikon Eclipse Ni-E wide field microscope with a Hamamatsu C11440 digital camera and 100x/1.45NA oil immersion objective. Identical exposure conditions were used at each wavelength for each pair of mock- and RAP-treated samples under comparison. Images were processed using Fiji software.

## Parasite growth and invasion assays

To determine the growth capability of mutant parasites over multiple cycles, DMSO and RAP-treated cultures were adjusted to a parasitaemia of 0.1%, with samples for flow cytometry being sampled for fixation every 48 h for up to 6 erythrocytic growth cycles. Pelleted parasites were fixed in 4% paraformaldehyde (Electron Microscopy Sciences) and 0.02% glutaraldehyde (Sigma-Aldrich) in PBS for 1 h at 37˚C, before being diluted into PBS. When ready for flow cytometry, samples were stained with a 1:10,000 dilution of Hoechst 33342 (ThermoFisher Scientific) in PBS for 30 min at 37˚C. The invasive capacity of parasites was determined by adding RBCs to highly synchronous Percoll-purified mature schizonts to obtain a parasitaemia of 1%. After further incubation for 4 h, cultures were fixed and the percentage of newly infected RBCs was determined. Parasitaemia and DNA replication were indicated by Hoechst-staining intensity determined using a LSRFortessa (BD Biosciences) flow cytometer, collecting a minimum of 10$^5$ events per sample. Data were analysed using FlowJo software. The relative plaque-forming capacity of parasites was determined as previously described [46, 70]. All experiments

were performed in triplicate using blood from different donors, and statistical analysis was performed using GraphPad Prism.

### Time-lapse video microscopy

To visualise merozoite egress, schizont-stage mock- or RAP-treated parasites were imaged as previously described [4, 69]. Invasion videos were similarly performed using schizonts purified from mock- or RAP- treated cultures mixed with fresh uninfected RBCs. DIC images were acquired every 150 ms and Nikon NIS Elements AR analysis software was used to produce the resulting time-lapse videos.

### Transmission electron microscopy

Schizonts of *RAMAloxP* parasites were harvested ~44 h following mock- or RAP-treatment, before being fixed in 2.5% glutaraldehyde and 4% formaldehyde in 0.1 M phosphate buffer. After fixation, cells were washed in 0.1 M phosphate buffer, embedded in 2% agarose and processed as described previously [4].

### Statistical analysis of TEM images

For each sample, three sections were taken from each of three of the 1 mm$^3$ blocks of agarose-embedded schizonts. Twenty schizonts from each block were quantified, scoring the number of rhoptries within each intracellular merozoite. Each rhoptry was also visually classified as being circular or club-shaped (i.e. possessing distinct neck and bulb regions). Statistical analyses of the data were performed using R 3.3.1 (https://www.R-project.org/) (R Core Team, 2013). Generalised linear models were fit using the glm() function, with the first model (rhoptry_count~DMSO_or_Rapa_status, family = Poisson) being used to test for a difference in rhoptry counts. For comparison of 'club-shaped' counts on the non-zero rhoptry data, the model used was club_count~DMSO_or_Rapa_status + rhoptry_count, family = Poisson. Analysis of Deviance was performed using the anova() function on the resultant glm object, with test = 'Chisq'.

### Transmission electron tomography

Tilt series images of a schizont from each experimental condition were collected from +60° to -60° with 1° increments using the Tecnai User Interface software (FEI). The 1024 x 1024 pixel images were recorded with an Ultrascan charge coupled device camera (Gatan Inc.) with a pixel size of 5.42 nm. The IMOD package [71] was used to reconstruct individual tomograms, with patch tracking used to create a fiducial model. For each schizont, tomograms from three serial 200 nm sections were flattened and joined together in z to obtain a continuous volume. The 3dmod programme of IMOD was used to manually segment rhoptry membranes, from which three-dimensional surface models were generated. Models were left open where rhoptries extended outside the joined volume, and where the top or bottom surface of a rhoptry was lost between adjacent tomograms.

## Supporting information

**S1 Fig. No significant differences in replication rates of *RAMAloxP P. falciparum* clones and parental 1G5DC parasites over the course of 4 erythrocytic cycles.** Parasitaemia values (quantified by flow cytometry) were averaged from three biological replicate experiments performed using blood from different donors, and are presented as mean ± SD.
(TIF)

**S2 Fig. Disruption of RAMA has no effect on subcellular localisation of the rhoptry neck protein RON12 or the rhoptry body and neck protein Rh2b.** IFA showing that the staining profiles of both RON12 and Rh2b were unaltered between DMSO- and RAP-treated *RAMA-loxP* clone 9C10 schizonts. Antibodies to RhopH2 were used as a marker for the rhoptry bulb. Scale bar, 5 μm.
(TIF)

**S1 Table. Oligonucleotide primer sequences used in this study.**
(PDF)

**S1 Movie. Time-lapse video microscopy showing that mock-treated (DMSO-treated) *RAMAloxP* clone 9C10 merozoites bind to and deform RBCs, invade and induce echinocytosis.**
(MP4)

**S2 Movie. Time-lapse video microscopy showing that RAP-treated, RAMA-deficient *RAMAloxP* clone 9C10 merozoites interact with and deform host RBCs but neither invade nor induce echinocytosis.**
(MP4)

**S3 Movie. Tomogram video and three-dimensional reconstruction of rhoptries from a mock-treated (DMSO-treated) *RAMAloxP* clone 9C10 schizont (see Materials and methods for a description of methods used).**
(MOV)

**S4 Movie. Tomogram video and three-dimensional reconstruction of rhoptries from of a RAP-treated *RAMAloxP* clone 9C10 schizont (see Materials and methods for a description of methods used).**
(MOV)

## Acknowledgments

The authors would like to extend special thanks to Ross Coppel for extensive discussions and the generous gift of anti-RAMA-D antibodies. They are also indebted to Fiona Hackett for invaluable support in generating and maintaining the DiCre-expressing parent lines, and thank Judith Green and Tony Holder for the generous donation of antibody reagents.

## Author Contributions

**Conceptualization:** Emma S. Sherling, Louis H. Miller, Michael J. Blackman.

**Data curation:** Matthew R. G. Russell.

**Formal analysis:** Emma S. Sherling, Abigail J. Perrin, Ellen Knuepfer, Matthew R. G. Russell, Lucy M. Collinson.

**Funding acquisition:** Emma S. Sherling, Louis H. Miller, Michael J. Blackman.

**Investigation:** Emma S. Sherling, Abigail J. Perrin, Ellen Knuepfer, Matthew R. G. Russell.

**Methodology:** Abigail J. Perrin, Ellen Knuepfer, Matthew R. G. Russell, Lucy M. Collinson.

**Supervision:** Lucy M. Collinson, Louis H. Miller, Michael J. Blackman.

**Validation:** Abigail J. Perrin.

**Visualization:** Abigail J. Perrin.

**Writing – original draft:** Emma S. Sherling, Abigail J. Perrin, Michael J. Blackman.

**Writing – review & editing:** Emma S. Sherling, Ellen Knuepfer, Louis H. Miller, Michael J. Blackman.

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
