## [Decision Letter · Decision Letter 0]

8 Jul 2019

Dear Mike

Thank you very much for submitting your manuscript "The Plasmodium falciparum rhoptry bulb protein RAMA plays an essential role in rhoptry neck morphogenesis and host red blood cell invasion" (PPATHOGENS-D-19-00959) for review by PLOS Pathogens. Your manuscript was fully evaluated at the editorial level and by independent peer reviewers. The reviewers appreciated the attention to an important topic but identified some aspects of the manuscript that should be improved.

We therefore ask you to modify the manuscript according to the review recommendations before we can consider your manuscript for acceptance. Your revisions should address the specific points made by each reviewer. Please pay particular attention to the additional work recommended by reviewer 3 regarding the localisation of additional rhoptry neck marker (RON12, PfRh1, PfRh2a and PfRh2b) to support any general statement as to the localisation of neck proteins in the RAMA mutant.

(1) A letter containing a detailed list of your responses to the review comments and a description of the changes you have made in the manuscript. Please note while forming your response, if your article is accepted, you may have the opportunity to make the peer review history publicly available. The record will include editor decision letters (with reviews) and your responses to reviewer comments. If eligible, we will contact you to opt in or out.

(2) Two versions of the manuscript: one with either highlights or tracked changes denoting where the text has been changed; the other a clean version (uploaded as the manuscript file).

We hope to receive your revised manuscript within 60 days or less. If you anticipate any delay in its return, we ask that you let us know the expected resubmission date by replying to this email.

[LINK]

Sincerely,

Oliver Billker

Associate Editor

PLOS Pathogens

Vern Carruthers

Section Editor

PLOS Pathogens

Kasturi Haldar

Editor-in-Chief

PLOS Pathogens

orcid.org/0000-0001-5065-158X

Grant McFadden

Editor-in-Chief

PLOS Pathogens

orcid.org/0000-0002-2556-3526

Reviewer's Responses to Questions

**Part I - Summary**

Reviewer #1: Apical organelles of merozoites play important roles during host cell invasion. The rhoptries are club-shaped structures with a bulb region that tapers to a narrow neck. Different proteins localize to the rhoptry bulb and neck, but the function of many of these proteins and how they are spatially segregated within the rhoptries is unknown. Using conditional disruption of the gene encoding the GPI-anchored rhoptry bulb protein, rhoptry-associated membrane antigen (RAMA), the Authors demonstrated that RAMA is indispensable for blood stage parasite survival. Contrary to previous reports, RAMA is not required for trafficking of rhoptry bulb proteins. Instead, RAMA-null parasites display selective mislocalization of rhoptry neck proteins (RONs) and produce dysmorphic rhoptries that lack a distinct neck region. The mutant parasites undergo normal intracellular development and egress but display significant defect in invasion and do not induce echinocytosis in target red blood cells. These results clearly proved an important role for RAMA in formation of the rhoptry neck structure. All the works were carefully designed, clearly presented, and the manuscript is well written. I have only a few comments for the improvement of this manuscript.

Reviewer #2: Apicomplexan parasites have specialized apical organelles such as rhoptries to enable successful invasion into red blood cells. This manuscript by Sherling et al uses a conditional knockdown of RAMA, a rhoptry bulb protein, to understand its contribution to the invasion process. The main conclusions of this paper are that knockdown of RAMA results in a defect in parasite invasion (but not egress), the loss of expression/localization of RON proteins (but not RAP1, AMA1 etc) and malformed rhoptries (from bulb shaped to a more circular morphology).

In general, this paper is very thorough in the presentation of the results and the characterization of the knockdown phenotype. For example, in the generation of the knockdown lines and subsequent phenotyping, the authors use two independent clones and characterize the knockdown in the presence of rapamycin using PCR to show the excision of the relevant genomic locus, Western blotting and IFA to show that the levels of expression are dramatically reduced. While phenotyping the invasion defect, the authors also use 1G5DC which is a non-relevant knockdown as a control. In addition, this paper uses live imaging, IFA and TEM to define that the defect was in invasion and not egress, that RON proteins are missing and that the rhoptries are more likely to not retain a bulb shaped morphology in the knock down lines. In addition, the Discussion is balanced in its interpretation and in the scope of what has been analysed. For example, the knockdown construct that is generated still expresses ~ 220 amino acids at the N -terminus and the authors provide an explaination based on previous results how this residual domain will not interact with complex partners and is no longer membrane-associated. In summary, all relevant controls have been included and a diverse array of tools used to characterize the knockdown phenotype.

My minor recommendations are:

1. While the characterization of egress, early and late schizont development is complete, it would have been good to have more characterization during attachment in the invasion process. This would involve using isolated merozoites and imaging during the time of attachment, which is challenging. Nevertheless, the main question that could be addressed is: What happens to RAP2, PfRh5, RhopH2/H3, AMA1 when a RAMA-knockdown merozoite contacts a red blood cell (or a select visualisation of some of the proteins)? Do these parasite ligands show surface release onto the red blood cell surface rather than targeted single foci? It would be interesting to observe if the defect to these invasion ligands is similar to the phenotype seen in Figure 6B of Riglar et al., Cell Host & Microbe 2011. The main reason for extending the characterization to merozoites and invasion is that you would be able to further define a potential additional defect in the RAMA-knockdown; that these invasion ligands that are present are also not being “expulsed” in the correct fashion, potentially due to the malformed rhoptries. As the paper currently stands, your main conclusion is a relationship between the lack of RONs and malformed rhoptries.

2. Invasion ligands are released from the rhoptries and micronemes during invasion. Would it be possible to analyse invasion supernatants for a combination EBAs (micronemal) and PfRhs (rhoptries), to see if these ligands are still released? Or what happens to AMA1? The main rationale for requesting these experiments are similar to the one above, which is to extend how malformed rhoptries contribute to a further defect in invasion separate from RONs that may extend to timing of release and processing of invasion ligands.

3. First paragraph of Introduction:

i) “five species of the genus Plasmodium” should be changed to “six species…”.

ii) “Invasion is a stringently orchestrated process”, the word stringently should probably be removed as there is significant variation in times of invasion.

4. The legend in Figure 1F suggests that there is quantitative data for the staining of RAMA in untreated and treated cells. Please include a graph of the anti-RAMA and anti-RAP1 staining in the WT and knockdown cells.

Reviewer #3: This manuscript assesses the function of the rhoptry protein RAMA in Plasmodium falciparum using conditional disruption of the corresponding gene. The authors show that the rhoptry-associated membrane protein RAMA is indispensable for the invasion process, likely because the rhoptry are not formed properly and not secreted. Previous studies using FRET and immunoprecipitations had shown that RAMA interacts with the rhoptry proteins RAP1 and RhopH3 and serves as an escorter for their proper targeting to the bulb of the rhoptries. Here the authors show that the apical targeting of these particular proteins is unchanged in absence of RAMA, invalidating this presumed function. In contrast the authors provide evidence that in absence of RAMA, RON2, RON4, and RON3 are not associated with mature rhoptries. Finally, insight into the ultrastructural consequences of depleting RAMA revealed its crucial role in the pear-shaped structure of the rhoptry. The wide bulb region remains formed but the elongated neck is missing or abnormal in absence of RAMA.

The study is very well performed and the paper very well written.

How two distinct compartments, the neck versus the bulb, are generated despite the absence ofa any physical barrier remains enigmatic in Apicomplexa. To my knowledge this is the first rhoptry protein described to participate in the formation of the neck of rhoptries in Plasmodium (but also in Apicomplexa in general), without affecting the morphology of the bulb part. This is an important step to move forward into elucidating the mechanistic of the elongation of rhoptry neck.

**Part II – Major Issues: Key Experiments Required for Acceptance**

Reviewer #1: Not applicable.

Reviewer #2: (No Response)

Reviewer #3: 1. One of the claim of this paper is that RAMA is required for selective trafficking of rhoptry neck proteins, while targeting of bulb proteins is not affected by absence of RAMA. While it is obvious that a subset of rhoptry proteins are not properly targeted, it is not clear whether the defect is selective for rhoptry neck proteins as claimed throughout the manuscript. The authors used too few rhoptry neck markers to generalize their findings. First, two proteins are part of the same complex (RON2, RON4) and second, despite its name, RON3 has actually been shown to be a rhoptry bulb protein in both Plasmodium (Ito et al., Parasitol Int, 2011) and Toxoplasma (Peter’s Bradley annotation in ToxoDB). The targeting defect appears therefore not selective for rhoptry neck proteins but also for bulb proteins. Furthermore, conflicting localization of PfRh5 are reported in the literature; in some papers PfRh5 is described as a rhoptry neck (Baum et al., Int J Parasitol 2009), or as a bulb protein (Sony Reddy, PNAS 2015) or in neither of the two (Douglas, J. Immunol. 2014). In summary, in order to get a better picture of the selectivity of RAMA-dependent targeting, the authors are recommended to perform IFAs using additional validated markers for rhoptry neck proteins (RON12, PfRh1, PfRh2a and PfRh2b).

2. In absence of RAMA, RON2 RON3 and RON4 proteins are visible in immature schizonts and then disappear in fully segmented parasites. This raises the question of whether they are in immature rhoptries together with rhoptry proteins RAP2, ARO, RhopH...; or if they are stacked in the secretory pathway and then degraded? A co-localization between RON2 (or RON4) and RAP2 (or ARO) will clarify if the block occurs before the proteins reach the rhoptries or once they are in the secretory organelles.

**Part III – Minor Issues: Editorial and Data Presentation Modifications**

Reviewer #1: Major Comment:

1. RON3 <page 18=""> & Fig. 4B&C

“Our observation that trafficking of RON3 was impaired in the RAMAΔE2 mutants is intriguing since in fact the Plasmodium orthologue of this protein has been previously localised to the rhoptry bulb [61], rather than to the rhoptry neck as originally determined for the Toxoplasma orthologue [10]; the current nomenclature for RON3 in Plasmodium may therefore be somewhat misleading.”

The original localization of TgRON3 was determined by IFA alone (not IEM) [10], so that the evidence of the naming as TgRON3 is weak. In contrast, since localization of PfRON3 in the rhoptry bulb was proved by IEM [61], PfRON3 as a rhoptry “bulb” protein is the only reliable localization evidence to date. So PfRON3 should be treated as rhoptry “bulb” protein NOT the rhoptry “neck” protein. Please revise the related text throughout the manuscript. Especially all the IFA images of the RON3 (Fig. 4B&C) are similar to those of RON2 & RON4 (rhoptry “neck” protein).

Minor Comments:

2. Fig. 4A

Rh5 signal in DMSO control parasite looks weaker (RAP parasite looks stronger). Current description is “n difference”. So, please explain.

3. Page 4 Lines 21-23.

“Egress and invasion involve the regulated discharge of at least four classes of secretory organelles, unique to apicomplexan parasites, that positioned within the apical end of the merozoite.”

Rhoptry, microneme, dense granule and what? Please describe 4 classes of secretory organelles with citation.

4. IFA method (page 23)

Please describe the dilution of each antibodies (“appropriate dilution” is not preferable).

5. Statistical analysis of TEM images (page 24-25)

“the glm() function”, “the anova() function”

What are these mean?

Reviewer #2: Please refer to the summary above.

Reviewer #3: 3. Of note, deletion of TgCA_RP in Toxoplasma also induces an unusual shape of the rhoptries (Chasen et al., MSphere 2017), although the phenotype is quite different, and the morphologies of the rhoptries are also quite different between the two Apicomplexa. Interestingly, TgCA_RP is also a GPI-anchored rhoptry protein associated with lipid-raft and its role in maturation of rhoptries is dependent of its GPI anchor. While no homolog of PfRAMA could be identified in T. gondii and RAMA and CARP do not share homologies, their overall mechanism might still be quite similar. This mutant should be introduced in the discussion.

4. Since this paper does not agree with the previous escort function of RAMA for HMW and LMW rhoptry proteins, immuno EM to obtain a full validation of the presence of RAP1 and RhopH3 in the rhoptry bulbs would have been enlightening but is not required for the central conclusions of the paper.

5. In discussion the authors noted that their findings may not necessarily be in conflict with a role of RAMA in facilitating formation of the HMW and LMW complexes. This is something that might be easy to test by co-IP using anti-RhopH3 or RAP2 antibodies in RAPA treated parasites.

6. Figure 4C, despite the reviewer agrees with the conclusion drawn from this panel, the pictures are too small to distinguish between immature schizonts versus fully segmented schizonts. The reviewer suggests to show only representative schizonts in larger magnification in main figure and to move the full fig. 4C as a supplementary figure.

7. Unless I am mistaken, the number of rhoptries analyzed for the figures 5 is not mentioned.

8. Page 10. What does “t =115.9, d.f.=4” mean?

 </page>

PLOS authors have the option to publish the peer review history of their article (what does this mean?). If published, this will include your full peer review and any attached files.

Reviewer #1: No

Reviewer #2: No

Reviewer #3: No

---

## [Decision Letter · Decision Letter 1]

27 Aug 2019

[EXSCINDED]

Dear Professor Blackman,

We are pleased to inform that your manuscript, "The Plasmodium falciparum rhoptry bulb protein RAMA plays an essential role in rhoptry neck morphogenesis and host red blood cell invasion", has been editorially accepted for publication at PLOS Pathogens. 

Before your manuscript can be formally accepted and sent to production, you will need to complete our formatting changes, which you will receive by email within a week. Please note that your manuscript will not be scheduled for publication until you have made the required changes.

IMPORTANT NOTES

(1) Please note, once your paper is accepted, an uncorrected proof of your manuscript will be published online ahead of the final version, unless you’ve already opted out via the online submission form. If, for any reason, you do not want an earlier version of your manuscript published online or are unsure if you have already indicated as such, please let the journal staff know immediately at plospathogens@plos.org.

(2) Copyediting and Proofreading: The corresponding author will receive a typeset proof for review, to ensure errors have not been introduced during production. Please review the PDF proof of your manuscript carefully, as this is the last chance to correct any errors. Please note that major changes, or those which affect the scientific understanding of the work, will likely cause delays to the publication date of your manuscript. 

(3) Appropriate Figure Files: Please remove all name and figure # text from your figure files. Please also take this time to check that your figures are of high resolution, which will improve the readbility of your figures and help expedite your manuscript's publication. Please note that figures must have been originally created at 300dpi or higher. Do not manually increase the resolution of your files. For instructions on how to properly obtain high quality images, please review our Figure Guidelines, with examples at: http://journals.plos.org/plospathogens/s/figures.

(4) Striking Image: Please upload a striking still image to accompany your article if one is available (you can include a new image or an existing one from within your manuscript). Should your paper be accepted, this image will be considered for our monthly issue image and may also appear on our website to feature your article. Please upload this as a separate file, selecting "striking image" as the file type upon upload. Please also include a separate "Other" file with a caption, including credits and any potential copyright information. Please do not include the caption in the main article file. If your image is from someone other than yourself, please ensure that the artist has read and agreed to the terms and conditions of the Creative Commons Attribution License at http://journals.plos.org/plospathogens/s/content-license. Please note that PLOS cannot publish copyrighted images.

(5) Press Release or Related Media: If your institution or institutions have a press office, please notify them about your upcoming paper at this point, to enable them to help maximize its impact. If they will be preparing press materials for this manuscript, please inform our press team in advance at plospathogens@plos.org as soon as possible. We ask that you contact us within one week to plan ahead of our fast Production schedule. If you need to know your paper's publication date for related media purposes, you must coordinate with our press team, and your manuscript will remain under a strict press embargo until the publication date and time. This means an early version of your manuscript will not be published ahead of your final version. 

(6)  PLOS requires an ORCID iD for all corresponding authors on papers submitted after December 6th, 2016. Please ensure that you have an ORCID iD and that it is validated in Editorial Manager.  To do this, go to ‘Update my Information’ (in the upper left-hand corner of the main menu), and click on the Fetch/Validate link next to the ORCID field.  This will take you to the ORCID site and allow you to create a new iD or authenticate a pre-existing iD in Editorial Manager

(7) Update your Profile Information: Now that your manuscript has been provisionally accepted, please log into Editorial Manager and update your profile, if needed. Go to https://www.editorialmanager.com/ppathogens, log in, and click on the "Update My Information" link at the top of the page. Please update your user information to ensure an efficient production and billing process. 

(8) LaTeX users only: Our staff will ask you to upload a TEX file in addition to the PDF before the paper can be sent to typesetting, so please carefully review our Latex Guidelines http://journals.plos.org/plospathogens/s/latex in the meantime.

(9) If you have associated protocols in protocols.io, please ensure that you make them public before publication to guarantee immediate access to the methodological details.

Best regards,

Oliver Billker

Associate Editor

PLOS Pathogens

Vern Carruthers

Section Editor

PLOS Pathogens

Kasturi Haldar

Editor-in-Chief

PLOS Pathogens

orcid.org/0000-0001-5065-158X

Grant McFadden

Editor-in-Chief

PLOS Pathogens

orcid.org/0000-0002-2556-3526

Reviewer Comments (if any, and for reference):

Reviewer's Responses to Questions

Part I - Summary

Reviewer #3: In their revised MS, the authors have addressed my comments from the initial reviews. In particular they have addressed properly my major point about the selectivity of RAMA-dependent targeting and the text has been revised throughout accordingly.

Part II – Major Issues: Key Experiments Required for Acceptance

Please use this section to detail the key new experiments or modifications of existing experiments that should be 

absolutely

 required to validate study conclusions.

Reviewer #3: (No Response)

Part III – Minor Issues: Editorial and Data Presentation Modifications

Reviewer #3: While this is not required for the central conclusions of the paper, the reviewer would like to clarify his “minor” comment 5. The suggestion was not to confirm the interaction between RAMA and the HMW and LMW complexes, but to perform IP using anti RAP2 antibodies in RAMAloxP parasites (+/- rapa) to test if in absence of RAMA, RAP2 protein remains in complex with RAP1 and RAP3. Indeed, as they discussed “RAMA may associate with RhopH3 and RAP1, for example, in order to facilitate formation of the respective HMW and LMW complexes.”

PLOS authors have the option to publish the peer review history of their article (what does this mean?). If published, this will include your full peer review and any attached files.

Do you want your identity to be public for this peer review?

 For information about this choice, including consent withdrawal, please see our Privacy Policy.

Reviewer #3: No

---

## [Editor Report · Acceptance letter]

30 Aug 2019

Dear Professor Blackman,

We are delighted to inform you that your manuscript, "The Plasmodium falciparum rhoptry bulb protein RAMA plays an essential role in rhoptry neck morphogenesis and host red blood cell invasion," has been formally accepted for publication in PLOS Pathogens.

Best regards,

Kasturi Haldar

Editor-in-Chief

PLOS Pathogens

orcid.org/0000-0001-5065-158X

Grant McFadden

Editor-in-Chief

PLOS Pathogens

orcid.org/0000-0002-2556-3526